# Vein Network and Climatic Factors Predict the Leaf Economic Spectrum of Desert Plants in Xinjiang, China

**DOI:** 10.3390/plants12030581

**Published:** 2023-01-28

**Authors:** Yi Du, Yulin Zhang, Zichun Guo, Zhihao Zhang, Fanjiang Zeng

**Affiliations:** 1Xinjiang Key Laboratory of Desert Plant Roots Ecology and Vegetation Restoration, Xinjiang Institute of Ecology and Geography, Chinese Academy of Sciences, Urumqi 830011, China; 2State Key Laboratory of Desert and Oasis Ecology, Xinjiang Institute of Ecology and Geography, Chinese Academy of Sciences, Urumqi 830011, China; 3Cele National Station of Observation and Research for Desert-Grassland Ecosystems, Cele 848300, China; 4University of Chinese Academy of Sciences, Beijing 100049, China; 5College of Ecology and Environmental, Xinjiang University, Urumqi 830046, China; 6State Key Laboratory of Soil and Sustainable Agriculture, Institute of Soil Science, Chinese Academy of Sciences, Nanjing 210008, China

**Keywords:** leaf economic spectrum, desert plants, adaptation strategy, vein network, climatic factors

## Abstract

The leaf economic spectrum (LES) has been repeatedly verified with regional and global datasets. However, the LES of desert plants and its drivers has not been fully explored at the species level. In this study, we sampled three desert perennial plant species (*Alhagi sparsifolia*, *Karelinia caspia*, and *Apocynum venetum*) at three different geographical areas of distribution in Xinjiang, China, and measured 10 leaf economic traits to determine their strategy of resource utilization. The scores of the first axis from the principal component analysis of 10 leaf economic traits as a continuous variable define the LES. This study showed that the LES did exist in desert plants in this region. The leaf economic spectrum shifted from a more conservative strategy to a more acquisitive strategy with increasing contents of soil potassium (K) and the ratio of K to phosphorus. Except for the vein density of *A. venetum,* which quadratically correlated with LES, the vein density, distance between veins, and vein loopiness significantly positively correlated with the LES (*p* < 0.05), indicating a covariation and tradeoff relationship. The annual mean temperature was significantly negatively correlated with LES, while the annual mean precipitation (MAP) and the aridity index (AI), which was calculated by the ratio of MAP to potential evapotranspiration, significantly positively correlated with the LES. Of these, vein loopiness and AI were more effective at predicting the change in LES from anatomical and climatic perspectives owing to their high regression coefficients (R^2^). The findings of this study will substantially improve the understanding of the strategies of desert plants to utilize resources and predict the structure and function of ecosystems.

## 1. Introduction

A series of covariant chemical, structural, and physiological attributes of leaves, such as the leaf mass per area, photosynthetic assimilation rates, contents of leaf nitrogen (N) and phosphorus (P), dark respiration rate, and leaf lifespan, constitute the leaf economic spectrum (LES), which was first proposed by Wright et al. on a global scale [1]. The LES sheds light on the effect of tradeoff strategies on leaf construction between fast-growing (i.e., acquisitive strategy) and slow-growing (i.e., conservative strategy) species [1,2]. One end of the LES embodies fast-return leaves, which have a high content of N, a high photosynthetic rate, and tend to have short life spans and low construction costs. The other end characterizes slow-return leaves, which have long life spans and a high construction cost but tend to have a low leaf N content and photosynthetic rates. In the ongoing process of the “investment-return” of plant leaves, there is a tradeoff between these leaf economic traits, and the correlation between these leaf economic traits has also been repeatedly verified at different scales, including individual [3,4], community [5], and global levels [6,7].

Although the LES has effectively characterized the variation of plant ecological strategies, the formation mechanism and driving forces across different study scales have not been consistently identified. Climate, biogeographic limitation, and genetic variation may contribute to the variation in leaf traits. Moles et al. evaluated the relationship between 21 plant traits of 447,961 species and climate factors worldwide and found that the influence of mean annual temperature (MAT) on the leaf functional traits was more significant than that of the mean annual precipitation (MAP) [6]. However, these two factors weakly predicted the variation of 14 leaf traits of 1103 individuals across species [8]. Leaf vein traits are closely related to plant performance and are considered to be the determinants of LES [9]. For example, vein density accurately predicts the change in photosynthetic assimilation rates because water and carbon transport in leaves are determined by the ability of water to pass through the veins [10]. Vein loopiness is the number of closed loops per unit area. Higher loopiness means more redundant routes for water and nutrients, reducing the risk of supply disruptions [9]. Despite the large number of cases and theoretical studies that have clearly described the driving factors of LES, the influence of climate factors and vein network on the LES of desert plants remains unclear.

Current knowledge on the LES is mainly tested from the dataset from forests and grasslands [1,2,4,6]. Factors influencing the adaptation strategies of species from desert ecosystems should largely come from underground (water and soil nutrients) rather than above ground (light) [11]. However, it is not clear whether the existing theories are applicable to desert species as well. Perennial desert species are an ideal system for exploring this question because their growth is constantly limited by water [12]. Here, we proposed the following hypotheses: (1) perennial desert plants in arid areas also follow the LES law, and more conservative traits would be observed in more arid climates, and vice versa, more acquisitive traits would be observed and (2) climatic factors (e.g., temperature and precipitation) and vein traits can effectively explain the variation in LES across these plants from the climate and vein network perspective, respectively, owing to the differences in adaptation among species.

To test these two hypotheses, we selected three perennial herbs, *Alhagi sparsifolia* (Fabaceae)*, Karelinia caspia* (Compositae), and *Apocynum venetum* (Apocynaceae), which are widely distributed in the three basins of Xinjiang, China. Ten leaf traits—leaf mass per area (*LMA*), carbon assimilation rates (i.e., maximum photosynthetic rate) on area (*A*_a_) and mass basis (*A*_m_), leaf nitrogen content on area (*N*_a_) and mass basis (*N*_m_), leaf phosphorus content on area (*P*_a_) and mass basis (*P*_m_), dark respiration rate on area (*R*d_a_) and mass basis (*R*d_m_), and photosynthetic nitrogen use efficiency (*PNUE*)—were assessed to determine the compatibility of LES with desert plants. These leaf traits were then correlated with soil properties, vein network properties, and climatic factors to explore the effect of predicting factors on the variation in LES. These experiments are an important complement and validation of the LES framework.

## 2. Results

The soil physical and chemical properties of the three sampling sites differed significantly (*p* < 0.05) (Table 1). Site Turpan had the highest TP content and EC, and the lowest K:P and pH. Site Cele had the lowest levels of SOC and TN. Site Mosuowan had a higher TK content, K:P, and lower TP content.

The first two axes of the principal component cumulatively explained 83.26% of the total variation for the 10 leaf economic traits (Figure 1). The significantly correlated relationship among these traits suggests the existence of an LES (*p* < 0.05) (Appendix A). The *LMA* and *N*_a_ of the three species were the lowest in site Mosuowan, while the *R*d_m_ and *PNUE* of three species were the highest at this site (Appendix A). The carbon assimilation rate significantly correlated with the dark respiration rate and N concentration on the basis of leaf area or mass. The leaf traits that were associated with a conservative strategy, such as a high *LMA*, had negative scores on the first PCA axis, while the traits associated with an acquisition strategy, such as a high *PNUE*, had positive scores on the first PCA axis (Appendix A).

Site (R^2^ = 0.540, *p* < 0.001) exerted a stronger effect than plant species (R^2^ = 0.225, *p* < 0.001) on the attributes of leaf economic traits (Figure 1). Sites Cele and Turpan with a lower content of soil K and K:P were linked with the conservative side of PCA, such as the high *LMA*, and thus, had the lowest scores (*p* < 0.05, Figure 2 and Figure 3). Conversely, site Mosuowan had the highest content of soil TK and K:P, which was associated with the more acquisitive side, such as high *A*_m_ and *PNUE*, and thus, had the highest scores (*p* < 0.05). The PCA first axis scores of *K. caspia* that grew in Cele and Turpan were comparable (*p* > 0.05), which could be related to the similar level of TK in the soil at both sites (*p* > 0.05) (Table 1). The content of soil TK and K:P were the indices that significantly positively correlated with the scores of the first axis of leaf economic traits of three species (*p* < 0.001) (Figure 3 and Appendix A).

Subsequently, the influence of vein characteristics, climate factors, and leaf economic characteristics was explored. The vein density and distance between the veins of *A. venetum* differed significantly among the three sites (*p* < 0.05) (Appendix A). The vein loopiness of *A. sparsifolia* and *K. caspia* differed significantly across the three sites (*p* < 0.05). Across the three sites, except for the leaf vein density of *A. venetum*, there was a significantly positive relationship between vein density, distance between veins, vein loopiness, and leaf economic traits (*p* < 0.05) (Figure 4). The leaf vein density of *A. venetum* showed a quadratic relationship with its leaf economic traits (Figure 4j). The regression coefficient (R^2^) of the relationship between vein loopiness and PC1 scores of the LES (R^2^ = 0.518–0.926) was higher than that of the other vein traits. The leaf vein traits significantly positively correlated with the *A*_m_ and *PNUE*, while they significantly negatively correlated with the *LMA* (*p* < 0.01) (Appendix A).

The MAP and AI exerted a significant positive relation with the leaf economic traits, while the MAT significantly negatively correlated with the leaf economic traits (Figure 5). The regression coefficient (R^2^) of these three climate factors on the PC1 scores of the LES of three plant species was AI (R^2^ = 0.530–0.967) > MAP (R^2^ = 0.498–0.956) > MAT (R^2^ = 0.390–0.904).

## 3. Discussion

This study revealed that the LES did exist in the three desert perennial species in different geographical distribution areas in Xinjiang, China, and the LES shifted from a more acquisitive to a more conservative strategy, which supports the first hypothesis of this study. Differences in the conditions of soil nutrients mediated by changes in the content of K and K:P contributed to the variation of LES. The leaf vein network, including vein density, vein loopiness, distance between veins, and climate factors, including MAT, MAP, and AI, exerted a powerful influence on the LES. Of these, vein loopiness and AI effectively predicted the variation in LES, which supports the second hypothesis.

Whether plants adopt more acquisitive or more conservative strategies depends on the availability of soil nutrients, which is ultimately reflected by the leaf economic traits [1]. The *LMA* is a hub trait in the LES. Plant species that grow on nutrient-poor soil tend to adopt a conservative strategy, and their leaves had a higher *LMA*. Species with a higher *LMA* can more effectively adapt to drought environments [7,13]. In contrast, plant species that grow on nutrient-rich soil adopt a more acquisitive strategy. A higher *PNUE* is associated with a lower *LMA* [14]. In this study, the *LMA* and *PNUE* in the first axis of PCA of 10 leaf economic traits traversed in two opposite directions and more effectively indicated more conservative and more acquisitive strategies, respectively (Figure 1 and Figure 2). Similar to the research of Delpiano et al. [5], this study found that the shift in three species from a conservative strategy to a more acquisitive strategy was caused by the increase in the contents of soil nutrients. Sites Cele and Turpan had a lower TK and K:P in the soil, and the species that grew at these sites exhibited the most conservative strategy (Table 1; Figure 2 and Figure 3). Conversely, in site Mosuowan, where K was less limiting owing to the higher TK and K:P, the species that grew there had more acquisitive strategies, such as a higher *PNUE* (Figure 3 and Appendix A). Higher nitrogen use efficiency (NUE) could contribute to the quicker growth and high productivity of plant species in site Mosuowan [15]. Although P and K are typically found in large quantities in soil, they are derived from rocks, and most of them are not easily absorbed by plants [16,17]. Manipulation experiments have revealed that the addition of P can improve the growth of plants characterized by conservative resource strategies [18], and the improvement in soil K induced the plants to exhibit the characteristics of resource acquisition [19]. These results suggest that the resource utilization strategies of these desert plants are primarily co-limited by soil K and P and can be predicted by these two nutrients in this region.

The leaf vein characteristics of these three desert plant species were effective predictors of the LES. Vein traits are closely interrelated with the gas exchange rate and fitness of plants since they determine photosynthesis, leaf nitrogen concentration, *LMA*, and other leaf economic traits [20]. Thus, it is important to identify the relationship between leaf vein network traits and LES and its ecological significance to predict the responses of plants and ecosystems to global change [21]. Plants with a higher leaf vein density have a higher carbon assimilation rate (*A*_m_) since the leaf water evapotranspiration and carbon acquisition are limited by the ability of leaf veins to transport water. In this study, the leaf vein density negatively correlated with the *LMA* and positively correlated with the *A*_m_ (Appendix A). In addition, the vein density, distance between veins, and vein loopiness were significantly positively correlated with the leaf economic traits, except for the vein density of *A. venetum* which quadratically correlated with the leaf economic traits (Figure 4). Interestingly, the PCA vectors for *A. venetum* were not strongly oriented along PC1 as they were in the other species (Figure 2), which suggests that the LES of this species is not as well represented by PC1. In turn, this may be related to why a quadratic relationship or generally weaker relationships between LES and soil properties, vein traits, and climatic factors were found in *A. venetum* compared to other species (Figure 2, Figure 3, Figure 4 and Figure 5). Overall, these results were consistent with the predictions of leaf vein theory [9]. Plant species at site Mosuowan had a higher vein density, distance between veins, and vein loopiness than those of the species at sites Cele and Turpan (Appendix A). A more effective leaf vein network conferred a higher *A*_m_ and *PNUE* for the plant species at site Mosuowan by improving the transport of leaf water per leaf area [9]. Thus, the species at site Mosuowan were characterized as having a more acquisitive strategy (Figure 4 and Appendix A). The amount of vein loopiness was highly accurate at predicting the LES because of its higher regression coefficient R^2^ (Figure 4). Higher loopiness provides more redundant routes for water and nutrient transport, which can reduce the risk of supply disruptions caused by a mechanical tear [9]. It is not surprising because the looped redundant structure of the vein system is important for improving the transport capacity of the network structure [22,23], although it increased the cost and consumption of leaves [24]. These results indicate that the leaf vein traits, particularly vein loopiness, can more effectively explain the variation of leaf economic traits of the plants in arid regions from physiological aspects.

Understanding the relationship between the LES and climate is the key to understanding plant adaptation to specific climatic regions and to predicting community and ecosystem responses to future climate change [25]. Previous research has shown that temperature and precipitation can strongly predict some plant functional traits with contrasting results. For example, at the global level, the concentrations of leaf N and P negatively correlate with temperature [26]. In contrast, Moles et al. [6] found that the level of leaf P positively correlates with temperature and that temperature is more effective than precipitation at indicating the changes in LES. In this study, the MAT, MAP, and AI significantly correlated with the leaf economic traits of the three desert species. Of these, AI was more accurate at predicting the change in leaf economic traits because of its higher regression coefficient R^2^ (Figure 5). This result is inconsistent with the results on a global scale, primarily because water is the key limiting factor for the survival of desert plants. Indeed, temperature indirectly affects the leaf traits of desert plant through water availability. Desert plants minimize evapotranspiration under high-temperature conditions by adjusting their morphology and physiology, such as by reducing their leaf area and stomatal conductance [4]. Increasing levels of drought (i.e., low AI) can increase the *LMA* and decrease the photosynthetic rate and NUE [27]. An increase in the *LMA* indicates an increase in water use efficiency (WUE) [4,28]. In this study, AI negatively correlated with *LMA* and *A*_a_, while it positively correlated with the *PNUE* (Appendix A), indicating that the plants that grow in this site with a high water supply (i.e., high AI) can enhance the *PNUE* against WUE. However, in the site with water deficiency, plants will improve their WUE and dry matter accumulation to extend leaf life and adapt to the adverse environment [4]. Thus, the AI can effectively explain the variation in leaf economic traits of desert plants in this region.

## 4. Conclusions

Overall, this study provided evidence that the changes in functional traits of three perennial desert species distributed in different geographical areas along environmental gradients represented a typical LES. Desert plants adapt to arid environments by a series of leaf economic traits, including adjusting the dry mass per area and *PNUE*, which is consistent with the shift from a conservative strategy to a more acquisitive strategy when environmental stresses are relieved. With the increase in the level of soil K and K:P, the LES shifted from more conservative to more acquisitive strategies. Vein density, the distance between veins, and vein loopiness were significantly positively correlated with the LES, indicating that leaf veins can effectively indicate the variation of LES from an anatomical perspective. Of these, the vein density of *A. venetum* quadratically correlated with the LES. Climate factors can also effectively predict the direction of LES variation. Thus, vein loopiness and AI more accurately predicted this direction and can be considered key factors to predict the adaption of desert plants to drought. In summary, this study highlights the important role of soil, vein networks, and climate in the coordination of LES for desert plants at the species level. The applicability of these factors to the community level merits further study.

## 5. Materials and Methods

### 5.1. Study Site Description and Sampling Design

The experiment was conducted at three field long-term in situ observation stations, Cele Desert Research Station (37°00′57″ N, 80°43′45″ E), Turpan Desert Botanical Garden (42°51′59″ N, 89°12′01″ E), and Mosuowan Desert Research Station (45°07′27″ N, 86°01′31″ E), Xinjiang Institute of Ecology and Geography, Chinese Academy of Sciences (XIEG, CAS) (Figure 6a). Of these, site Turpan had the highest MAT and lowest MAP. Site Mosuowan had the highest MAP, followed by Cele (Figure 6b). According to the description of the aridity index (AI) of the recent study [29], which is the ratio of MAP to mean annual potential evapotranspiration, sites Turpan and Cele are in the hyper-arid region (AI < 0.05), while Mosuowan is in the arid region. Most rainfall occurs in the summer (June to August) in the form of several bursts of rain, which are interspersed with droughts [30]. These climate data (2003–2013) were collected from the weather stations belonging to XIEG, CAS, about 1 km away from the sampling sites. In each sampling site, six 20 m × 20 m plots with a distance of 10 m were randomly established in the natural area of distribution of each plant species in July 2013. For each species, six morphologically similar plants (height and crown width were relatively consistent) among these plots (i.e., one plant per plot) were selected for leaf sampling.

### 5.2. Determination of the Leaf Economic Traits

We measured a set of 10 leaf morphological and physiological traits that are associated with the LES. Three to six mature leaves of similar size were chosen from each plant to obtain the leaf economic traits, including the *LMA*, *A*_a_, *A*_m_, *N*_a_, *N*_m_, *P*_a_, *P*_m_, *R*d_a_, *R*d_m_, and *PNUE*. The dust was wiped off the leaf surface, and a portable gas-exchange measurement system (LI-6400XT; Li-COR, Lincoln, NE, USA) was used to measure the instantaneous net photosynthetic rate of targeted leaves under different light conditions from 9:00 to 12:00 on a continuous sunny day, and the temperature, humidity, and concentrations of carbon dioxide (CO_2_) in the leaf chamber were controlled at 25 °C, 20%, and 400 μmol/mol, respectively. The maximum net photosynthetic rate (*A*_a_ and *A*_m_) and the dark respiration rate (*R*d_a_ and *R*d_m_) were obtained by fitting the light response curve with the pairwise data of net photosynthetic rate and light intensity that were obtained. These leaves were then scanned to assess the leaf area (CanoScan LiDE 110; Canon, Inc., Tokyo, Japan) and then processed with ImageJ software (1.52g; NIH, Bethesda, MD, USA). After that, the leaves were dried to a constant weight at 75 °C to calculate the mass. The oven-dried leaves were ground to powder, and their contents of N and P were determined by the Kjeldahl and molybdenum-antimony anti-colorimetric methods, respectively. The photosynthetic nitrogen use efficiency (*PNUE*) was estimated by the ratio of photosynthetic rate to N content per leaf area.

### 5.3. Determination of the Traits of Leaf Vein Network

The leaf vein traits were assessed by vein density, distance between the veins, and vein loopiness. We randomly selected 50 mature and healthy leaves from each species and placed them into a centrifuge tube that contained FAA fixation solution (5 mL of 37% formaldehyde, 5 mL of 100% glacial acetic acid, and 90 mL of 50% ethanol). These samples were rinsed with distilled water in the laboratory, and four pieces of the same area of tissue were cut from them (Figure 7). The mesophyll tissue, epidermis, and veins were dissociated by immersion in a 10~20% solution of NaOH. These samples were then immersed in distilled water, and the epidermis and mesophyll tissue were gently brushed away to obtain the veins. Then samples were bleached with 10% aqueous hydrogen peroxide until milky white, and they were then washed with distilled water and stained with safranin. The samples were then rinsed, dehydrated by a series of gradients of alcohol, flattened, sealed by paraffin, and photographed using a microscope (Olympus BX51; Olympus, Tokyo, Japan). Five photos were taken for each piece. The vein density, distance between veins, and vein loopiness in the photo were calculated by Image J software (Figure 7). Vein density was calculated by the total path length of the vein per unit area. Distance between vines was evaluated by the mean diameter of the largest circular masks that fit in each closed loop. Vein loopiness was measured by the number of closed loops per unit area [10].

### 5.4. Determination of the Soil Physical and Chemical Properties

We randomly collected five 0–100 cm soil samples along the diagonal of each quadrate, which were mixed, sifted through a 2 mm sieve, and air-dried to determine the physical and chemical properties. The soil organic carbon (SOC), total N (TN), total P (TP), total potassium (TK), electrical conductivity (EC), and pH were measured as described by Zhang et al. [31]. Briefly, the potassium dichromate oxidation method was used to measure the SOC content. The TN content was measured using the Kjeldahl method. The TP and TK contents were determined using inductively coupled plasma optical emission spectrometry (ICP-OES) (iCAP 6300; Thermo Fisher Scientific, Waltham, MA, USA). The soil EC and pH were measured in a 1:5 (*w*/*v*) mixture of soil and water using an EC meter (DDSJ-319L, Shanghai INESA Scientific Instrument Co. Ltd., Shanghai, China) and a pH meter (PHSJ-6L, INESA Scientific Instrument Co. Ltd.), respectively.

### 5.5. Statistical Analysis

All the statistical analyses were conducted in R version 4.2.2 [32]. One-way analyses of variance (ANOVAs) followed by Duncan’s method were performed to assess the difference in leaf economic traits across three geographical populations (sampling sites). To explore the independent and interactive effect of species and site on the leaf economic traits (Euclidean distance), a permutation multivariate analysis of variance (PERMANOVA) was conducted with 999 permutation tests using the adonis function in vegan package [33]. A principal component analysis (PCA) was used to visualize the effects of species and site on leaf economic traits, and their first axis scores were used to quantify the LES. The first axis scores from the PCAs were used as a proxy of the gradients of trait variation across sites because they explained a high proportion of the variation in these leaf traits as described by Delpiano et al. [5]. To evaluate the differences in leaf economic spectrum (i.e., resource use strategies) of the specific plant species across sites, one-way ANOVAs followed by Tukey HSD post-hoc tests were conducted to quantify the differences among sites using the first axis scores of PCA as dependent variables and site as the factor. The standardized major axis regressions were conducted to assess the degree of coordination between the leaf economic spectrum and vein network traits and climate factors. Heatmaps visualized the Pearson correlations between leaf economic characteristics and soil properties, vein networks, and climatic factors.

## Figures and Tables

**Figure 1 plants-12-00581-f001:**
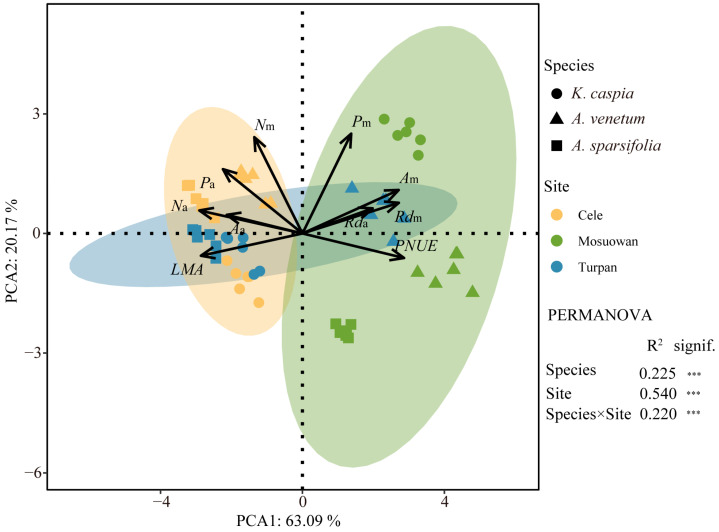
Principal components analysis (PCA) for ten leaf economic traits at the individual level at Cele, Turpan, and Mosuowan. The ellipse represents a 95% confidence interval. Permutation multivariate analysis of variance (PERMANOVA) with 999 permutation tests was used to explore the independent and interactive effects of plant species and sampling sites. R^2^. explained quantity; ***. *p* < 0.001. *LMA*. leaf mass per area; *A*_a_. carbon assimilation rates on area basis; *A*_m_. carbon assimilation rates on mass basis; *N*_a_. leaf nitrogen content on area basis; *N*_m_. leaf nitrogen content on mass basis; *P*_a_. leaf phosphorus content on area basis; *P*_m_. leaf phosphorus content on mass basis; *R*d_a_. dark respiration rate on area basis; *R*d_m_. dark respiration rate on mass basis; *PNUE*. photosynthetic nitrogen use efficiency.

**Figure 2 plants-12-00581-f002:**
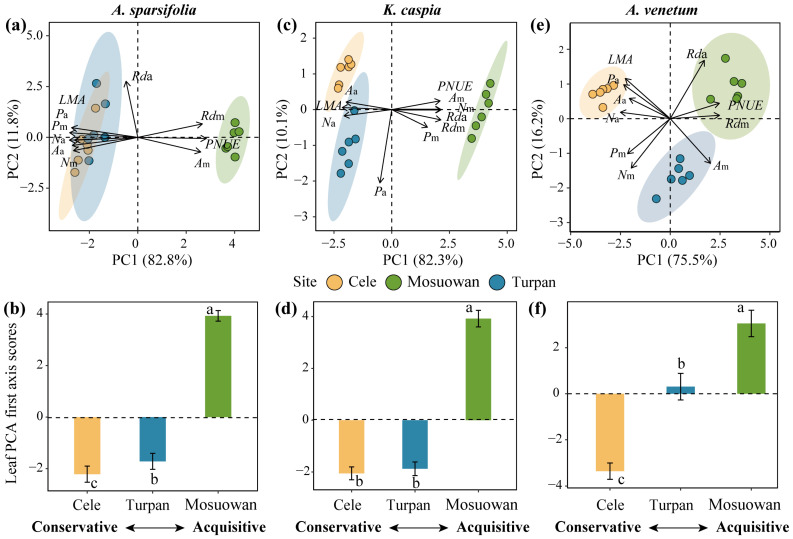
Principal components analysis (PCA) for ten leaf economic traits of *A. sparsifolia* (**a**), *K. caspia* (**c**), and *A. venetum* (**e**) at Cele, Turpan, and Mosuowan. The ellipse represents a 95% confidence interval. (**b**,**d**,**f**) mean (±SD) scores of the first axis (PC1) for three plant species at three sampling sites. Different lowercase letters indicate significant differences (*p* < 0.05) across different sampling points. *LMA*. leaf mass per area; *A*_a_. carbon assimilation rates on area basis; *A*_m_. carbon assimilation rates on mass basis; *N*_a_. leaf nitrogen content on area basis; *N*_m_. leaf nitrogen content on mass basis; *P*_a_. leaf phosphorus content on area basis; *P*_m_. leaf phosphorus content on mass basis; *R*d_a_. dark respiration rate on area basis; *R*d_m_. dark respiration rate on mass basis; *PNUE*. photosynthetic nitrogen use efficiency.

**Figure 3 plants-12-00581-f003:**
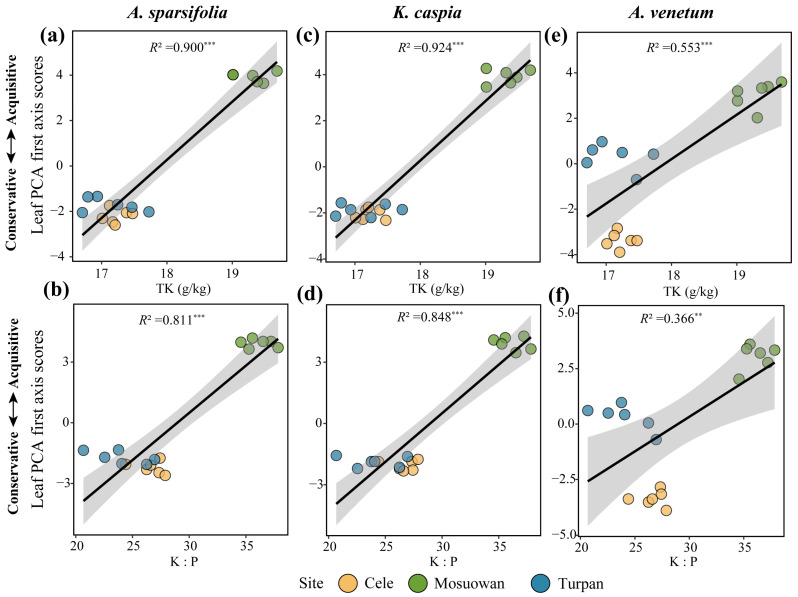
Liner regressions between leaf economic traits and soil total K (TK) content (**a**,**c**,**e**) and K:P ratio (**b**,**d**,**f**). **. *p* < 0.01; ***. *p* < 0.001.

**Figure 4 plants-12-00581-f004:**
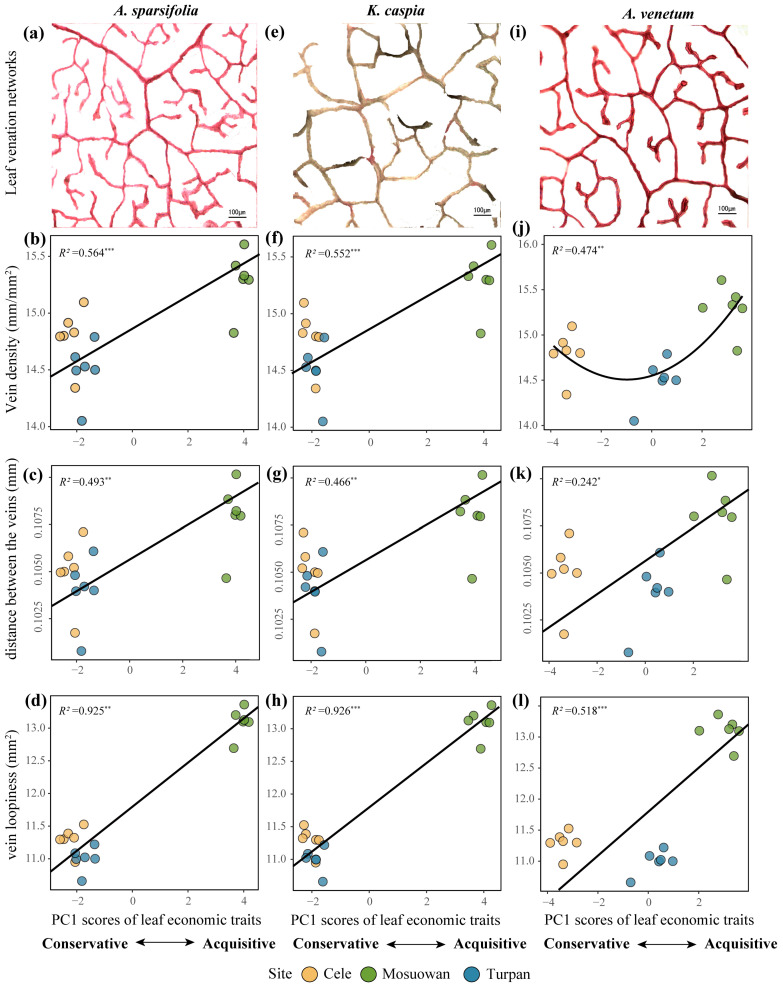
Leaf venation networks (**a**,**e**,**i**) and standardized major axis (SMA) regressions between the first axis scores of leaf economic traits and vein density (**b**,**f**,**j**), distance between veins (**c**,**g**,**k**), and of loopiness of veins (**d**,**h**,**l**) of three plant species. The black lines indicate significant SMA regression within three sampling sites. *. *p* < 0.05; **. *p* < 0.01; ***. *p* < 0.001.

**Figure 5 plants-12-00581-f005:**
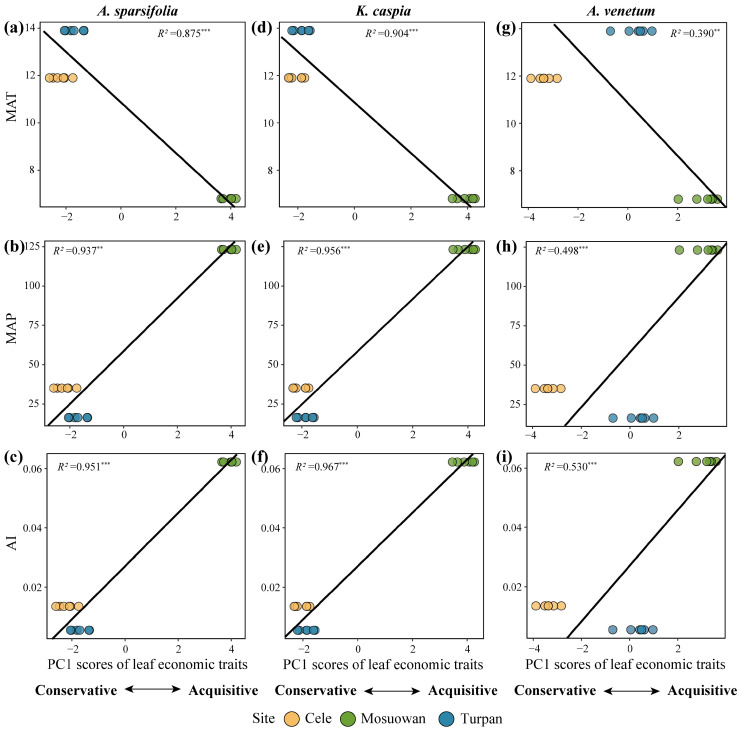
Standardized major axis (SMA) regressions between the first axis scores of leaf economic traits and mean annual temperature (MAT, (**a**,**d**,**g**)), mean annual precipitation (MAP, (**b**,**e**,**h**)), and aridity index (AI, (**c**,**f**,**i**)) of three plant species. The black lines indicate significant SMA regression within three sampling sites. **. *p* < 0.01; ***. *p* < 0.001.

**Figure 6 plants-12-00581-f006:**
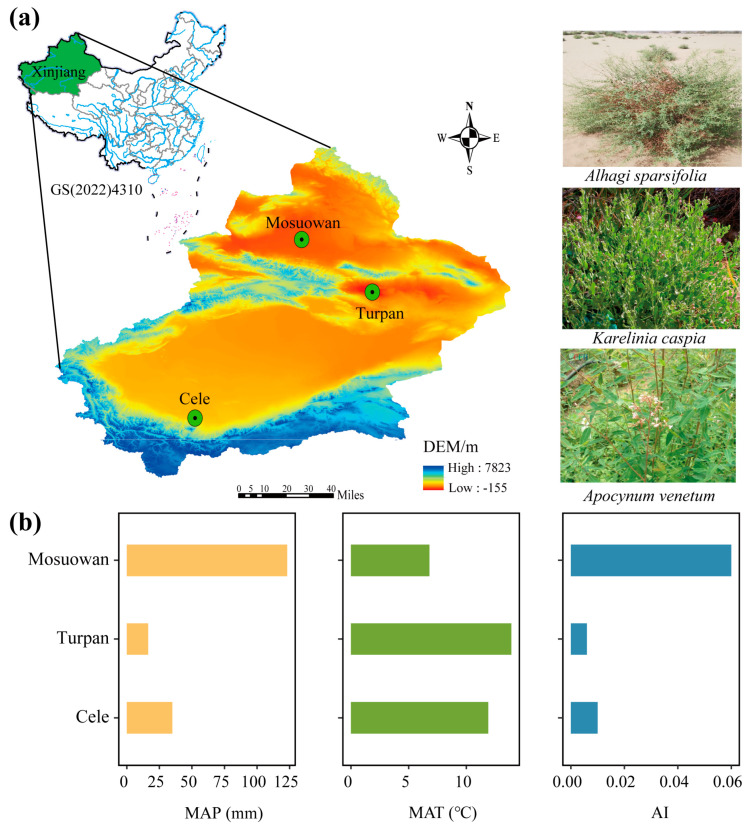
(**a**) The three sampling sites at Cele, Turpan, and Mosuowan located in Tarim Basin, Tur−pan Basin, and Junggar Basin, respectively. The sampling plants were *A. sparsifolia*, *K. caspia*, and *A. venetum*. (**b**) The climate factors of three sampling sites. MAT. mean annual temperature; MAP. mean annual precipitation; AI. aridity index.

**Figure 7 plants-12-00581-f007:**
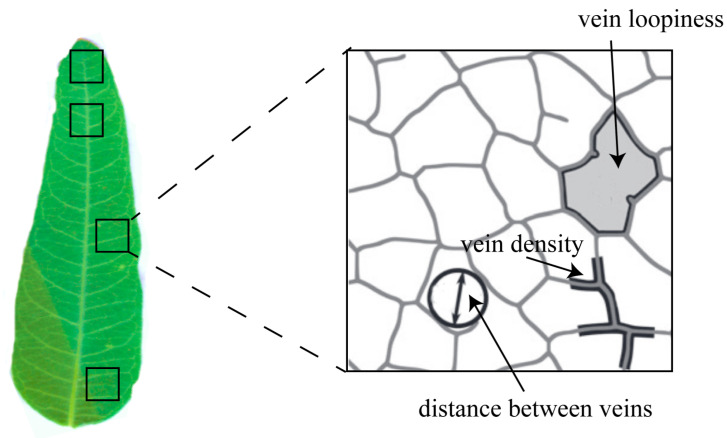
Leaf vein traits (Adapted from [10]).

**Table 1 plants-12-00581-t001:** Soil properties (mean ± SD) of each sampling site. Different lowercase letters indicate significant differences (*p* < 0.05) across different sampling points. SOC. Soil organic carbon; TN. total nitrogen; TP. total phosphorus; TK. total potassium; EC. electrical conductivity.

Soil Properties	Cele	Turpan	Mosuowan
SOC (g/kg)	1.70 ± 0.20 b	4.41 ± 1.23 a	3.65 ± 0.58 a
TN (g/kg)	0.13 ± 0.01 b	0.38 ± 0.10 a	0.28 ± 0.05 a
TP (g/kg)	0.65 ± 0.04 b	0.72 ± 0.07 a	0.53 ± 0.02 c
N:P	0.21 ± 0.01 b	0.53 ± 0.13 a	0.53 ± 0.07 a
TK (g/kg)	17.23 ± 0.17 b	17.14 ± 0.40 b	19.31 ± 0.27 a
K:P	26.65 ± 1.25 b	24.04 ± 2.32 c	36.16 ± 1.24 a
pH	8.87 ± 0.10 a	8.26 ± 0.11 b	8.89 ± 0.13 a
EC (μS/cm)	403.13 ± 76.40 b	2239.33 ± 727.55 a	141.57 ± 20.63 b

## Data Availability

The data presented in this study are available in the graphs and tables provided in the manuscript and Appendix A.

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
