# Peer review of "Vein Network and Climatic Factors Predict the Leaf Economic Spectrum of Desert Plants in Xinjiang, China"

_plants, 2023, doi:10.3390/plants12030581_

Round 1
Reviewer 1 Report
In this manuscript, Du et al present a study of the Leaf Economics Spectrum (LES) in three desert species in NW China. Using LES 10 traits (4 traits on both per-mass and per area bases plus two other traits) they fit PCAs to all the data and to each species separately and used the first PCA axis as the proxy for LES position, which they correlated against other traits included three measures of the vein network, three measures of climate and several measures of soil nutrients. I note that they measured 6 soil traits plus one synthetic trait but do not present them all in the results. The authors conclude that vein loopiness and aridity index are the best predictors of LES in these species based on having the highest R2 values when regressed with PC1. Overall I thought this was a good study with fairly clear hypotheses that are suitable to the data and well addressed by the data. Regarding the approach, on one hand I think it is useful to distill various traits into one PCA axis and use this to represent the LES. But on the other hand I have some doubts about the approach where it is concluded that a given trait predicts LES. In this study, each species has a different PCA with some differences in which vectors align strongly with PC1 and which do not. The difference in PCAs suggests that LES differs between species which suggests that vein loopiness, for example, would not predict LES in a novel species the same way it predicts LES in these three study species. Moving forward, can it be assumed that vein loopiness is a strong proxy for LES in any species? Or will it always be necessary to construct a 10-trait PCA to get PC1 for every species? Based on your findings, we could conclude that high vein loopiness correlates with a high PC1 score and therefore an acquisitive strategy, but we could not conclude that this indicates high Nm because the Nm vector has positive PC1 for K caspia and negative PC1 for A. sparsifolia. So I'd suggest some thought about the implications of using PC1 as a proxy for LES when PCAs differ between species.
Title: You state vein loopiness and AI "more effectively" predict LES. But you do not compare it to something else; i.e. more than what? Having read the abstract, it feels like the title is just one result chosen to be the title, rather than the title representing the study more broadly. It's also notable that the introduction does not explicitly mention vein loopiness or why it should be important. You mention vein density but otherwise only mention "vein traits" and "vein network". So it would be helpful to describe why the various vein traits, particularly loopiness, are important. In the discussion, the importance of the veins is mentioned with regard to the Sack et al and Blonder et al papers, and there is some mention of the functional consequences of vein density. But I did not see a discussion about why vein loopiness is functionally important or a definition of what it is.
18-19: This study isn't really looking at LES on the individual scale (i.e. variation within an individual) but is rather on a species scale (i.e. variation between individuals of a given species).
20: Please list all species as scientific names rather than one with common name. You use scientific names throughout and never refer to this as sword-leaf dogbane.
21-22: It seems you are essentially just saying the first axis of the PCA existed. A little rewording needed to link LES to PC1.
45: "leaves" stated twice
45: It's important to specify if N content is on a per area or per mass basis.
46,47: "life spans"
55-56: vein networks are a leaf trait so it should not be said they contribute to leaf trait variation
62: The paper you cite does not support this claim. In fact it rejects it. From the abstract "Careful re-examination of leaf anatomy, published datasets, and a newly compiled global database for diverse species did not support the ‘vein origin’ hypothesis"
71: The transition feels very abrupt between introducing desert climates and stating hypotheses. I would like to see a little more build up for why LES should be studied in desert plants. You cite the adverse conditions in arid climates but is there something about arid climates that should make us expect LES to differ in desert species vs other species? Similarly, you state that information on LES in desert species is largely unknown, but is it totally unknown? If we know something from the literature already, then what do we know? What do we still not know?
73: Can you make this first hypothesis a little more specific? The term "adaptation strategies" is vague and not obviously testable. Also stating that traits "differ in different areas" could be made stronger by stating how they should differ e.g. more conservative traits in more arid climates.
90: Your site names are all single words, so I don't find it necessary to abbreviate them, especially because you already have many abbreviations for trait names, including TP = total phosphorus and TP= Turpan.
99: Should say "three species" instead of "three plants".
102: It's curious that Nm has a negative PC1 value, suggesting it is part of the conservative strategy according to your interpretation of PC1.
146: Something to consider is that a trait may explain X% of PC1, that PC1 only explains Y% percent of the data. So the trait explains less than X% of the data.
168: The first hypothesis is only that LES exists - not that it's driven by nutrient availability.
190: Not clear what "these two" nutrients refers to because the previous two sentences mention N, P and K.
206: I don't follow regarding leaf mass being "certain" and LMA decreasing as VLA increases. Denser veins should increase LMA.
216: Are they categorically acquisitive or are they just more acquisitive than other sites?
239: Does LMA causatively affect WUE? Similarly at line 244 you imply that increasing WUE causatively increases leaf lifespan, but is that true or is it correlative?
246: It seems amiss to conclude with the suggested role of WUE when the study did not examine this trait.
259: It's notable that the PCA vectors for A venetum are not strongly oriented along PC1 as they are in the other species, which suggests that LES is not as well represented by PC1 in this species. In turn, this may be related to why a quadratic relationship and generally weaker relationships were found in this species compared to others.
Methods: I did not see it stated why these three species were chosen for the study. A brief description about their distribution, family and growth form (e.g. I'm not sure if they are herbaceous or woody) etc would be useful. Site coordinates and a little more description about where they are would also be useful. For example are these sites within reserves? rangelands? sand dunes? salt plains? floodplains? etc.
275: How near are the weather stations and do they have publicly available data and a website to cite here?
293: It would be better to use one symbol for net photosynthesis instead of Pn and A. Especially because you use P for phosphorous.
297: Not sure what you mean by ImageJ Pro 6.0. The most recent version of ImageJ is 1.54b.
312: This process of physically isolating the veins seems risky and prone to damaging or removing veins by accident. In the method I'm familiar with from the literature and from experience, the tissues are not separated and the veins are stained for visualization after chemically clearing the other tissues. In Fig 4, the veins look fairly intact but not separating the tissues is something to consider.
314: Dehydrated in alcohol? Also please state what you used to seal them.
316: Some more information is needed here regarding how distance between veins and vein loopiness were calculated
342: "Tukey"
Fig 2: Perhaps I missed something but why are there 6 points per species? Shouldn't there be one point per site*species combination (i.e. 3 points per species)?
Fig 3: Here you have PC1 on the y axis and in Fig 4 and 5 PC1 is on the x axis. Is there a reason for this? Also here you show confidence intervals but you do now show these in Fig 4 and 5.
Fig 4: Vein traits are missing units. You also have "lopiness" (missing an o).
Fig 6: Elevation does not seem relevant to show in the map. Showing a climatological variable such as AI seems it would be more relevant. Perhaps elevation is shown to indicate basin but 1) that's not readily apparent and 2) there's not explanation about why choosing from different basins was done. Also it would be good to state what the whole map is. E.g. is it a province? The main text says northwest China.
Fig 5: You have two panel "d"s and misplaced "PC1" in panel e and a misplaced "1" in panel h.
Fig 6: Here and elsewhere you have "Alhagi sparsifoliar" instead of "Alhagi sparsifolia"
Fig S1: You do not specify what the size of the squares represents.
Reviewer 2 Report
It is a well clearly written paper in an interesting issue, how plant traits evolve to adapt to different ambients like deserts. Nevertheless, I have seen some flaws in the description of the work within the methods section that have to be clarified. These are hghlited in the manuscript with yellow and notes.
